# Effects of Flight on Reproductive Development in Long-Winged Female Crickets (*Velarifictorus aspersus* Walker; Orthoptera: Gryllidae) with Differences in Flight Behavior

**DOI:** 10.3390/insects14010079

**Published:** 2023-01-12

**Authors:** Ye-Song Ren, Bin Zhang, Yang Zeng, Dao-Hong Zhu

**Affiliations:** Laboratory of Insect Behavior and Evolutionary Ecology, College of Life Science and Technology, Central South University of Forestry and Technology, Changsha 410004, China

**Keywords:** flight behavior, trade-off, flight muscle, reproduction, migration

## Abstract

**Simple Summary:**

In wing dimorphic insects, long-winged (LW) females were generally considered to be potential migrants, whereas short-winged females were considered sedentary. The aim of this research was to investigate the flight ability of LW females in the wing dimorphic crickets *Velarifictorus aspersus* and examine the effect of flight on ovarian development in LW females with different flight capacities. We provide the first evidence that flight ability varied among LW females, and the critical flight time for switching from flight to reproduction varied among LW *V. aspersus* female crickets with polymorphic flight behavior.

**Abstract:**

A trade-off between the capacity for flight and reproduction has been documented extensively in wing polymorphic female insects, thereby supporting the possible fitness gain due to flightlessness. However, most of these studies were conducted without considering the effect of flight behavior. In the present study, we assessed the flight duration by long-winged (LW) females in the cricket species *Velarifictorus aspersus* on different days after adult emergence and examined the effect of flight on ovarian development in LW females with different flight capacities. Our results showed that the flight capacity increased with age and peaked after 5 days. In addition, the flight capacity varied among individuals, where most LW females could only take short flights (sustained flight time < 10 min) and only a few individuals could take long flights (sustained flight time > 20 min). In LW female crickets demonstrating only short flights, repeated flying for 30 or 60 min significantly promoted reproductive development. However, in those capable of long flights, reproductive development was affected only after a flight of 60 min. The flight muscles degraded after the start of rapid reproduction in those with both short and long flights. Our results indicated that the critical flight time for switching from flight to reproduction varies among LW *V. aspersus* female crickets with polymorphic flight behavior.

## 1. Introduction

The acquisition of wings enabled insects to evolve the ability to fly, which greatly enhanced their adaptability, and it is one of the main reasons for the success of insects. However, flight polymorphism is common in many insect taxa [1]. Flight polymorphism can be characterized in terms of flight behavior and flight muscles, and wing polymorphism [2,3,4]. Flight behavior polymorphism has been reported in some winged monomorphic insect species [5,6]. Some individuals are migrants whereas others are non-migrants. Migrants and non-migrants are generally assessed based on the duration of a single continuous flight [5]. For example, McAnelly [5] proposed that *Melanoplus sanguinipes* individuals with continual flights for less than 10 min were non-migrants and those with uninterrupted flights for over 1 h were potential migrants. Wing polymorphism has been found in more than 10 insect taxa [2,7,8]. Some adults have fully developed wings and are capable of flying, whereas others have no wings or only degenerate residual wings and are unable to fly. Flight muscle polymorphism has been reported in some wing monomorphic and wing polymorphic insects [7,9,10]. For example, some long-winged (LW) females of the cricket *Gryllus firmus* have pink and developed flight muscles, whereas the flight muscles are degraded and milky white with no flight function in others [7]. LW females with white flight muscles are not able to fly and their reproductive organs are similar to those of SW females, whereas those with pink flight muscles can fly but their reproductive organs are smaller than those of SW females [11].

The ability to fly enhances the adaptability of insects but it also incurs huge energy costs, including for the development and maintenance of flight organs, and the synthesis of energy materials to sustain flight [12,13,14]. Losing the capacity for flight may increase the investment of energy in other life-history traits. The fitness cost of the capacity for flight has typically been assessed by comparing the decrease in the fecundity of LW females that have not flown with that of short-winged (SW) females in wing polymorphic insects [7]. Many studies have reported that SW females generally developed their ovaries more rapidly and laid a higher number of eggs compared to LW females, thereby suggesting a trade-off between the capability of flight and reproduction [15]. The trade-off between the capacity for flight and reproduction provides important evidence for the gain of fitness in the evolution of flightlessness. Because LW females will fly in the field, it is important to investigate the effect of flight behavior on reproduction for examination of whether this trade-off exists under natural conditions. However, the effect of flight behavior on the reproduction of LW females has been rarely studied, and contradictory results are given. For example, in the wing dimorphic cricket *Gryllus texensis*, a flight of 5 min can significantly promote reproduction in LW females [16]. However, for another wing dimorphic cricket species *Gryllus rubens*, a flight of 1 h can not increase the fecundity of LW females [17].

The cricket species *Velarifictorus aspersus* exhibits distinct wing dimorphism. A trade-off has been identified between the capacity for flight and reproduction in both female and male crickets when flight is restricted [18], but this effect may be attenuated when the flight time reaches a critical threshold [19]. Thus, in the present study, we investigated the flight duration of LW females of *V. aspersus* on different days after emergence to test whether flight ability varied within LW females. We then determined the effect of flight on ovarian development in LW females with differences in terms of their flight ability.

## 2. Materials and Methods

### 2.1. Experimental Crickets

*V. aspersus* crickets were obtained from an established laboratory colony that originated from dozens of crickets collected in 2010 in Hainan Province, China. About 100 adults were maintained for each generation of the laboratory population. We also collected the field crickets each year and reared them with the laboratory population. Crickets were reared with ad libitum access to food (corn powder, fresh potato, and cabbages) and water under a light: dark regime (LD) of 16:8 h and at 30 °C, as described by Zeng and Zhu [18]. Newly hatched nymphs were raised in groups (50 nymphs) in plastic containers (30 cm × 18 cm × 20 cm) under LD = 16:8 h and at 25 °C. After emergence, the adults were moved to another container (10 cm × 10 cm × 10 cm) and kept individually under the same conditions for subsequent experiments.

### 2.2. Investigation of Flight Ability of LW Females at Different Ages

A previous study showed that the wings of LW adults disappear 10 days after emergence [18], so we investigated the flight ability of LW females at 1, 3, 5, 7, and 9 days after emergence (40 crickets for each day). Crickets were stimulated to fly using the method described by Zeng et al. [19]. Briefly, LW crickets were glued at the pronotum to a wooden applicator stick and placed in front of a small fan to promote flight. The wind speed was about 2.5 m/s. Flight experiments were conducted during the first 4 h of night and the chamber was illuminated using a red light during the dark period. We observed that many crickets could fly for a short time on each occasion, but they then continued to fly after a rest. We just found out 2 min of rest may stimulate the cricket of this species to fly again in our preliminary observation. Therefore, when crickets stop flying, they were provided with a wooden stick to rest for 2 min, and flight experiments were ended when crickets did not fly in two successive tests. The total flight time and maximum continuous flight time were recorded for each LW female tested.

### 2.3. Effect of Flight on Reproductive Development

LW females aged 4–5 days after emergence were stimulated to fly 4 times to judge their flight ability. The next day crickets were stimulated to fly several times to reach a total flight time of 30 or 60 min. As a control, crickets were tethered so they could not fly: they were glued to a wooden applicator stick and provided with a wooden stick on which they rested instead of flying. Crickets were dissected at 0, 24, and 48 h after flight, and their flight muscles and ovaries were weighed.

### 2.4. Data Analysis

Data were presented as the mean ± standard error. Because data of the total flight time were not normally distributed (Kolmogorov–Smirnov test, *p* = 0.008), the data of total flight time were log-transformed and analyzed by one-way analysis of variance (ANOVA), followed by Tukey’s test. Effects of flight ability, flight time, and time after the flight on flight muscle or ovarian mass were analyzed by multi-way ANOVA. We also analyzed the effect of flight time on flight muscle and ovarian mass of short-fliers or long-fliers at each time point by one-way ANOVA, followed by Tukey’s test. All analyses were conducted using SPSS 13.0 software.

## 3. Results

### 3.1. Flight Behavior Polymorphism in LW Females

In the observation period of 4 h, the total flight time of the LW females varied significantly among different age groups (ANOVA, F_4, 195_ = 46.49, *p* < 0.001) (Figure 1). The total flight time of the LW females increased dramatically during the first 5 days (Tukey’s test, *p* < 0.05), and did not change significantly at 7 and 9 days (Tukey’s test, *p* > 0.05). These results indicated that the flight ability of LW females in *V. aspersus* was affected by age and that the flight ability reached maximum 5 days after emergence.

On the first day after emergence, all females flew for less than 10 min. However, on the third, fifth, seventh, and ninth days, the maximum single flight time varied substantially among individuals. About half of the tested insects could not fly longer than 10 min in a single flight, while some had a maximum flight time between 10 and 20 min, and others could fly longer than 20 min in a single flight (Figure 2). These results indicated that the flight ability varied among LW females, where most of the LW females could only take short flights and a small proportion could take long flights. Because most of those individuals that can continually fly for 20 min were able to fly much longer, we defined insects with a maximum single flight time of less than 10 min as short-fliers, and insects with a maximum single flight time of longer than 20 min as long-fliers. To test the consistency of the duration of the flight, 100 LW females aged 5–6 days after emergence were stimulated to fly four times. Individuals flying less than 10 min on all four flights were defined as short-fliers, and individuals flying longer than 20 min in a single flight were defined as long-fliers. The flight times of crickets were re-tested 24 h later, and all 51 of those classified as short fliers also flew for less than 10 min the next day, while 25 of 28 classified as long fliers had a flight of >20 min.

### 3.2. Effect of Flight on Ovarian Development in LW Females with Differences in Their Flight Ability

The ovarian mass was significantly affected by flight ability, flight time, and time after flight (Multi-way ANOVA, flight ability: F_1, 315_ = 4.54, *p* = 0.03; flight time: F_2, 315_ = 53.03, *p* < 0.001; time after flight: F_2, 315_ = 244.93, *p* < 0.001). The flight muscle was significantly affected by flight time and time after flight (Multi-way ANOVA, flight time: F_2, 315_ = 7.68, *p* = 0.001; time after flight: F_2, 315_ = 22.51, *p* < 0.001), and there was a significant interaction between flight ability and time after flight (F_2, 315_ = 3.17, *p* = 0.04). When the short-fliers were stimulated to fly for 30 or 60 min, their flight muscles were significantly degraded 48 h after the completion of the flight (ANOVA, F_2, 57_ = 31.47, *p* < 0.001), and the ovarian mass were significantly heavier than that of the unflown group at 24 or 48 h after the flight was completed (ANOVA, 24 h: F_2, 57_ = 16.28, *p* < 0.001; 48 h: F_2, 57_ = 49.62, *p* < 0.001) (Figure 3A,C). For the long-fliers, flight could also significantly affect flight muscle (ANOVA, 48 h: F_2, 42_ = 12.77, *p* < 0.001) and ovarian development (ANOVA, 24 h: F_2, 42_ = 7.97, *p* = 0.001; 48 h: F_2, 42_ = 24.53, *p* < 0.001). However, the flight muscles and ovaries of the long-fliers were not significantly different from the unflown group within 48 h after a flight of 30 min (Tukey’s test, unflown versus 30 min: *p* > 0.05). After a flight of 60 min, the flight muscles were significantly degraded at the 48 h (Tukey’s test, unflown versus 60 min: *p* < 0.001), the ovarian development was significantly promoted 24 h after the completion of the flight (Tukey’s test, unflown versus 60 min: *p* < 0.001) (Figure 3B,D). These results indicated that flight significantly affected ovary development and the flight muscles in LW females, and the critical flight time for the LW females to start rapid reproduction differed between short-fliers and long-fliers.

## 4. Discussion

In the present study, we found that the total flight duration of LW individuals increased markedly between the first and third days after emergence, reaching a peak at 5 days, suggesting that flight muscles require further development during the early stage of adulthood [18]. In some wing monomorphic insect species, newly emerged adults also require a period to further develop their flight muscles and prepare for migration [9,20].

The 1 h rule has been used to identify migrants and non-migrants in several studies [6,21,22]. However, we found that only 9 of the 120 individuals tested during 5–9 days after emergence could fly continuously for more than an hour (Figure 2). Many individuals flew for 100–150 min over 4 h which could take them substantial distances, distances that if a consistent spatial orientation is maintained could be considered migratory. By contrast, it is likely that those that fly continuously for less than 10 min take only local flights. This suggestion needs to be tested further using the recapture method in the field.

The different flight abilities of LW female *V. aspersus* crickets also reveal that flight behavior affects reproductive development differently. A flight of 30 or 60 min promoted ovarian development in the short-fliers, whereas only a flight of 60 min promoted ovarian development in the long-fliers. A previous study found that a flight of 5 min had no effect on reproductive development in LW female *V. aspersus* crickets [19]. These findings suggested that the flight time may serve as a signal in LW *V. aspersus* to switch from migration to reproduction, and long-fliers could need to fly longer than short-fliers to start rapid reproduction. In *Aphis fabae*, fresh-winged females would not settle down when placed on a host plant, but they became progressively more ready to stay, feed, and larviposit the longer that they had flown [23,24]. Graham [25] also found that flight had a similar effect on the striped ambrosia beetle *Trypodendron lineatum*. In the migratory grasshopper *M. sanguinipes*, the performance of a long-duration flight to voluntary cessation or exhaustion accelerated the onset of first reproduction and enhanced reproductive success over the entire lifetime of the insect [26]. These studies also demonstrated that flight time could play an important role in the activation of reproduction. In two other wing dimorphic cricket species, a flight of 5 min was able to promote reproduction in LW *G. texensis* [16], whereas a flight of more than 1 h was unable to promote the reproductive output by LW females in *G. rubens* [17]. *G. rubens* crickets can fly for up to 10 h in the laboratory, so this species may need to fly for much longer than 1 h before it settles down and starts rapid reproduction.

If the results that we obtained for *V. aspersus* are generally applicable to wing-polymorphic insects, then LW adults may be divided into local fliers and migrants. The local fliers may fly shortly to avoid predators or to search for food, mates, and new sites located nearby for oviposition [27]. The migrants may have the ability to explore new habitats and increase gene exchange between populations. In wing dimorphic cricket *G. firmus*, LW females biosynthesized a greater amount of total lipid and triglyceride, whereas SW females biosynthesized a greater amount of phospholipid and ovarian protein [28,29]. In locusts, carbohydrates are typically used at the initial stage of flight or during short-term flight. Conversely, long-term flight requires lipid oxidation [30]. Several studies reported that gregarious locusts that were able to fly constantly for a long time had higher lipid reserves than solitary locusts which only take short flights [31,32]. Whether this energy metabolic difference exists in LW females of *V. aspersus* needs to be further investigated. Peptide adipokinetic hormones (AKHs) are synthesized and stored by neurosecretory cells of the corpus cardiacum. In locusts, flight activity will trigger the release of AKHs, and the action of AKHs on their fat body target cells triggers a number of coordinated signal transduction processes which culminate in the mobilization of both carbohydrates and lipids [33]. Therefore, these hormones play an important role in regulating the transmission of flight fuels into flight muscles. The AKH content of the corpora cardiaca is higher but less effective in the mobilization of lipid reserve in solitary locusts than in gregarious ones [33]. In addition, the classical model of juvenile hormone (JH) control of dispersal and reproduction suggests that a low titer of JH promotes flight muscle development and flight capability maintenance, whereas a high titer of JH induces rapid reproductive development [7,34]. However, recent studies have shown that the JH titer in flight-incapable insects exhibited very little diel variation during early adulthood, whereas the JH titer in the flight-capable morph exhibited a 10- to 20-fold spike near the end of the photophase and the beginning of the scotophase on each of several days in early adulthood [35,36]. Therefore, it has been hypothesized that the short-term increase in the JH titer may regulate some aspects of flight behavior. However, this diel variation in JH was not found in flight-capable *M. sanguinipes* [26], possibly because an elevated JH titer is positively correlated with flight duration and the JH titer may be higher in migrants than local fliers. Although AKHs and JH are involved in the regulation of flight and reproduction of insects, their regulation mechanism is still unclear. We consider that further studies of the physiological differences between migrants and local fliers of LW *V. aspersus* may provide important evidence regarding the physiological mechanism related to the trade-off between flight and reproduction.

## Figures and Tables

**Figure 1 insects-14-00079-f001:**
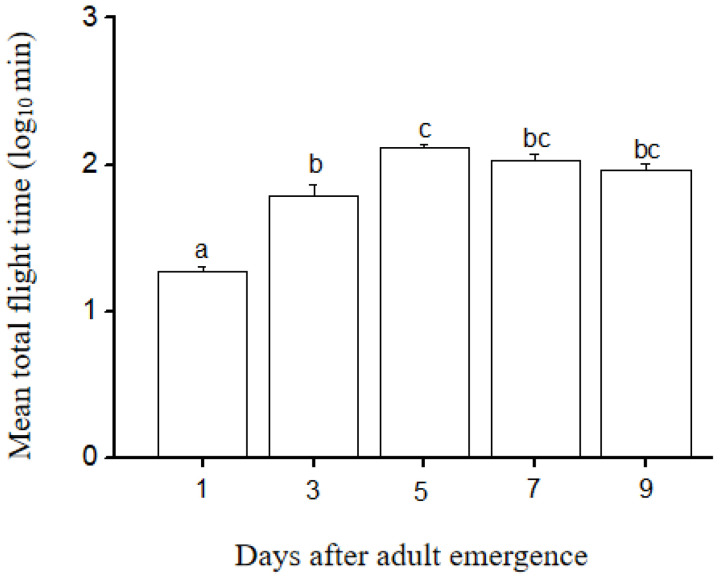
Effect of age on the mean total flight time of LW females of *V. asperses* stimulated to fly in a 4 h period. Different letters indicate a significant difference between groups by post hoc multiple comparisons of the means, *p* < 0.05, n = 40 for each group.

**Figure 2 insects-14-00079-f002:**
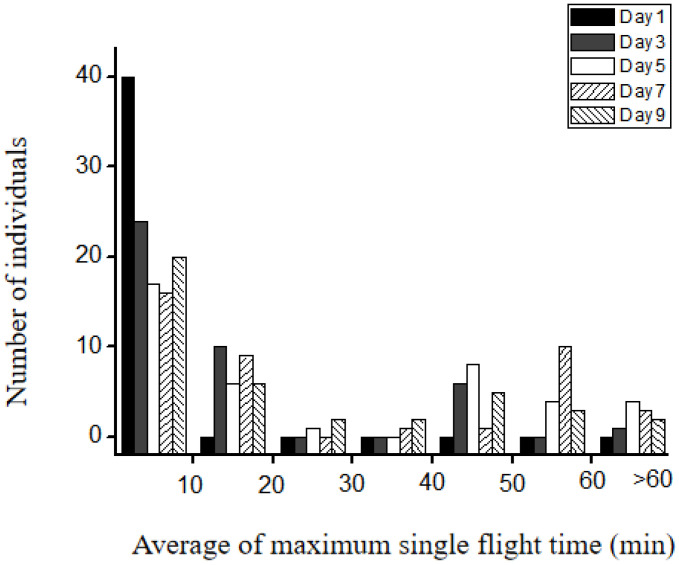
Distribution of the maximum of single flight among individuals on different days after emergence.

**Figure 3 insects-14-00079-f003:**
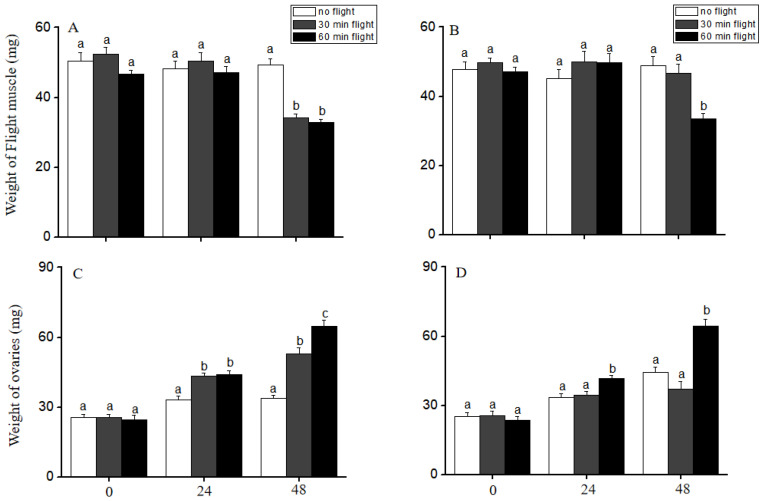
Effect of flight of 30 or 60 min on flight muscle and ovarian development in short-fliers (**A**,**C**) or long-fliers (**B**,**D**) of *V*. *aspersus*. Different letters indicate a significant difference between groups by ANOVA, post-hoc multiple comparisons of the means, *p* < 0.05, n = 20 for each group.

## Data Availability

The data that support the findings of this study are available on request from the corresponding author.

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
