# Peer review of "Effects of Flight on Reproductive Development in Long-Winged Female Crickets (Velarifictorus aspersus Walker; Orthoptera: Gryllidae) with Differences in Flight Behavior"

_insects, 2023, doi:10.3390/insects14010079_

Round 1
Reviewer 1 Report
A most interesting study that needs some revisions that to me fall somewhere between minor and major.

Author Response
Comments on Ren et al Flight and Reproductive development
General comment: A most interesting study that clearly shows that while short-wingedadults are relatively sedentary, some long winged adults are short-fliers that can disperse locally while others with long wings fly for much longer allowing the species to take advantage of potentially favourable habitats where they are, in nearby areas and some distance away. Some revisions to clarify some aspects are given below:
Specific comments:
Line 9: “long-winged (LW) adults were generally considered to be potential migrants, whereas short-winged adults were considered as sedentary ”
Answer: it has been revised, see in line 10.
Line 11: “in the wing dimorphic crickets ”
Answer: it has been revised, see in line 11.
Line 17: “ in wing polymorphic insects”
Answer: it has been revised, see in line 17.
Line 34: “insects to evolve the ability to fly”
Answer: it has been revised, see in line 34.
Line 89-90: You mention a few crickets not able to fly and had white flight muscles (like Gryllus firmus in line 46-47). However, why include this sentence at all since you do not mention flight muscle colour anywhere else: neither for long-fliers or short-fliers.
Answer: this sentence has been deleted.
2.3 Omit section on Definition of short and long fliers as it is a Result that you repeat in lines 135-136 of Results. I suggest adding some of this in lines 136+ of the Results.
Answer: this section has been deleted in the Material, and some of this has been added in the Results, see in lines 136-141.
2.4 now becomes 2.3 and is altered as follows:
Line 103+crickets were stimulated to fly several times to reach a total flight time of 30 or 60 min. As a control, crickets were tethered so they could not fly: they were glued to a wooden applicator stick and provided with a wooden stick on
which they rested instead of flying. Crickets were dissected at o, 24 o
Answer: it has been revised, see in lines 102-105.
Results
Line 120: “and did not change significantly”
Answer: it has been revised, see in line 119.
Figure 1 needs clarification:
In the Materials in line 88, it is stated that “flight time and maximum continuous flight time were recorded for each female”. change the X axis of fig. 1 to “mean total flight time”, and change the text to “Effect of age on the mean total flight time of LW females of V. aspersus stimulated to fly 4 times in a 4 h period. Different letters indicate a significant difference However, how were the total flight times over the 4 h period obtained? Were the females stimulated to fly 4 times in a 4 h period as stated in Line 95, or were the flight experiments ended when crickets stopped flying after two successive rests as stated in lines 87-88?
Answer: the X axis has been changed to “mean total flight time”, and the text has been changed to “Effect of age on the mean total flight time of LW females of V. asperses stimulated to fly in a 4 h period. Different letters indicate a significant difference between groups by ANOVA, p < 0.05, n = 40 for each group.” The total flight times were obtained when crickets did not fly in two successive tests.
In Figure 1 & hereafter, no need to repeat genus name: use V. aspersus as you do in line 121
Answer: it has been changed as “ V. aspersus”
Line 126: “ On the first day after emergence, all females flew for less than 10 min. However, on the third, fifth, seventh and ninth days, the maximum single flight time among individuals. About half of the tested insects could not fly longer than 10 min in a single flight, while some had a maximum flight time between 10 and 20 min, and others could fly longer than 20 min ...”
Answer: It has been revised accordingly, see in lines 126-130.
Line 136+ : “20 min as long-fliers. To test the consistency of duration of flight, the flight times of crickets were re-tested 24 h later: all 51 of those classified as short fliers also flew for less than 10 min the next day, while 25 of 28 classified as long fliers had a flight of >20 min.”
Answer: It has been revised accordingly, see in lines 136-141.
There seems to be an inconsistency between the number of insects tested (Line 95: 100 individuals aged 5-6 days tested, Line 184: 200 tested) compared to what is in Figure 2. If I add together the day 1 individuals, I get 40 at <10 min flight at perhaps another 10 in the longer fliers and also at day 5, I again get about 50 individuals: 18 short fliers, 7 flew 10-20 min and about 2, 1, 8, 5 and 6 in the >20-minute classes. How many were tested?
Answer: there is no inconsistency, and each 40 crickets were tested at each day in fig.2. The scale of X axis starts from -2, so it looks like there are more than 40 crickets tested for each day. Actually, for day 1, all 40 crickets fly less than 10 min, and no crickets fly longer than 10 min. Another 100 crickets aged 5-6 days were tested for consistency of flight duration, and these crickets were not included in fig.2.
Section 3.2 of results needs re writing. It would be clearer if for all statements, you put flight muscle effects first because they are first in the Figures.
Answer: section 3.2 has been rewrote according to this comment, see in lines 147-159.
Discussion
Line 166: “Among wing polymorphic insects, SW individuals can not fly while LW individuals can, but the actual flight abilities of the latter can vary greatly. In the present study, we found that the total flight duration of LW individuals increased markedly between the first and third days after emergence, reaching a peak at 5 days, suggesting that flight muscles require further development during the early stage of adulthood [15]. In some”
Answer: it has been revised accordingly, see in 171-175.
So replace all that of lines 180-193 with something like this:
This 1 h rule has been used to identify migrants and non-migrants in several studies [5, 18,19]. However, we found that while only 10 of the 200 individuals tested could fly continuously for more than an hour (Fig. 2), many individuals flew for 100-150 minutes over a 4 h period which could take them substantial distances, distances that could be considered migratory. By contrast, it is likely that those that fly continuously for less than 10 min take only local flights.
Answer: it has been revised accordingly, see in 184-190.
Reviewer 2 Report
The authors focused on the flight polymorphism among the long-winged types, even among the wing polymorphisms, and examined the variation in their muscle mass and ovarian development in this interesting study. This study is suitable for publication in 'insects' but needs significant modifications as indicated below.
Title
L2-4. This study was conducted on LW only, and the current title reads as if both SW and LW were examined and compared; it does not convey the content of this study that there is polymorphism in flight ability within LW females and that muscle mass and weight of ovaries vary and trade-offs exist within LW females.
Intoroduction
L33. The originality of this paper is not just the study of the trade-off between flight and reproduction, but L13-15, which focused on LW and the diversity within it, right?
L33. Please read “RG Harrison (1980) Annual Review of Ecology and Systematics, 11:95-118.”
L36. Please include appropriate citation.
Materials & Methods
L70. What is the information on wing polymorphism in the field in this species? Wing polymorphism and density are very closely related (Fujisaki (1986) RPE 28 43-52.), and in this experiment, is rearing density? Could there be an effect of high density? Could this be a reflection of the field? How many generations are kept at what density? How does the progressive rearing effect this experiment?
L73. What did they feed on?
L79. … adults disappear at 10 days……
L87. Why ‘2 min.’?
L91. The division into short fliers and long fliers is arbitrary and difficult to understand, since there is no obvious binomial distribution. Why not analyze the time of flight as a continuous variable? Or, why don't you apply a statistical distribution to see if they can be divided into two?
L106. What about wind speeds and other information?
L113. First, short and long are compared, and then the relationship between time after fly and effect of flight duration is analyzed, so the analysis should be conducted by two-way analysis of variance or GLM(M) instead of one-way analysis of variance.
Results
L131. It is very hard to see. Y-axis should be Average of maximum single flight time (min.), right?
Discussion
L165. The significance of the diversity within the LW of this species in the field and the discussion of physiological mechanisms in grasshoppers (other than JH, which has already been discussed) are needed.
Please be cautious in discussing whether or not to fly, as you should be discussing data within the LW and not the SW.
Differential lipid biosynthesis underlies a tradeoff between reproduction and flight capability in a wing-polymorphic cricket
Zhangwu Zhao and Anthony J. ZeraAuthors Info & Affiliations
December 16, 2002
99 (26) 16829-16834
https://doi.org/10.1073/pnas.262533999
Intermediary metabolism and life-history trade-offs: differential metabolism of amino acids underlies the dispersal-reproduction trade-off in a wing-polymorphic cricket
AJ Zera, Z Zhao - The American Naturalist
L166-167. Delete. Authors already mentioned.
L204. What is ‘larviposit’?
L205-206. striped ambrosia beetle
Author Response
The authors focused on the flight polymorphism among the long-winged types, even among the wing polymorphisms, and examined the variation in their muscle mass and ovarian development in this interesting study. This study is suitable for publication in 'insects' but needs significant modifications as indicated below.
Title
L2-4. This study was conducted on LW only, and the current title reads as if both SW and LW were examined and compared; it does not convey the content of this study that there is polymorphism in flight ability within LW females and that muscle mass and weight of ovaries vary and trade-offs exist within LW females.
Answer: the title has been revised.
Intoroduction
L33. The originality of this paper is not just the study of the trade-off between flight and reproduction, but L13-15, which focused on LW and the diversity within it, right?
Please read “RG Harrison (1980) Annual Review of Ecology and Systematics, 11:95-118.”
Answer: we focused on LW and the diversity within it.
L36. Please include appropriate citation.
Answer: reference has been added, see in line 36.
Materials & Methods
L70. What is the information on wing polymorphism in the field in this species? Wing polymorphism and density are very closely related (Fujisaki (1986) RPE 28 43-52.), and in this experiment, is rearing density? Could there be an effect of high density? Could this be a reflection of the field? How many generations are kept at what density? How does the progressive rearing effect this experiment?
Answer: we collected both LW and SW adults of this species in the field, demonstrating that this species is wing dimorphic in the field. However, we have not investigated the wing polymorphism of this species in the field. Our previous study showed that high density condition will induce LW adults, so we used a high rearing density to obtain more LW adults for this experiment. Although high rearing density will affect wing differentiation, and we have reared this species for more than 10 generations in the Lab, but we collected new crickets from field to mix with the Lab population.
L73. What did they feed on?
Answer: crickets feed on corn powder, fresh potato and vegetables, see in line 81.
L79. … adults disappear at 10 days……
Answer: it has been revised as “disappear”, see in line 88.
L87. Why ‘2 min.’?
Answer: Based on our observation, a rest of 2 min is enough for crickets to fly again
L91. The division into short fliers and long fliers is arbitrary and difficult to understand, since there is no obvious binomial distribution. Why not analyze the time of flight as a continuous variable? Or, why don't you apply a statistical distribution to see if they can be divided into two?
Answer: In this study, we divide shot flyer and long flyer according to their maximum single flight time.
L106. What about wind speeds and other information?
Answer: the wing speed is about 2.5 m/s, see in line 92.
L113. First, short and long are compared, and then the relationship between time after fly and effect of flight duration is analyzed, so the analysis should be conducted by two-way analysis of variance or GLM(M) instead of one-way analysis of variance.
Answer: we did not compare short and long flyers, and we think it is suitable to use ANOVA to the effect of flight time on flight muscle and ovarian mass of short-fliers or long-fliers at each time point.
Results
L131. It is very hard to see. Y-axis should be Average of maximum single flight time (min.), right?
Answer: it has been revised as “Average of maximum single flight time” , see in fig. 2
Discussion
L165. The significance of the diversity within the LW of this species in the field and the discussion of physiological mechanisms in grasshoppers (other than JH, which has already been discussed) are needed.
Please be cautious in discussing whether or not to fly, as you should be discussing data within the LW and not the SW.
Differential lipid biosynthesis underlies a tradeoff between reproduction and flight capability in a wing-polymorphic cricket
Zhangwu Zhao and Anthony J. ZeraAuthors Info & Affiliations
December 16, 2002
99 (26) 16829-16834
https://doi.org/10.1073/pnas.262533999
Intermediary metabolism and life-history trade-offs: differential metabolism of amino acids underlies the dispersal-reproduction trade-off in a wing-polymorphic cricket
AJ Zera, Z Zhao - The American Naturalist
Answer: energy metabolic and AKH regulation have been discussed, see in lines 214-231.
L166-167. Delete. Authors already mentioned.
Answer: it has been revised.
L204. What is ‘larviposit’?
Answer: production of larvae
L205-206. striped ambrosia beetle
Answer: it has been revised as “striped ambrosia beetle”, see in lines 201-202.
Reviewer 3 Report
Interaction between flight and reproductive development is a topic that have attracted much research attention in flying animals, and specifically in insects. Some important evolutionary aspects were suggested in cases where the insect species demonstrate flight-related polymorphism (morphological or behavioral). Different cricket species have been previously reported to show wing dimorphism – i.e., short vs. long wings morphs. The interactions between this dimorphism and aspects of the insects’ reproductive biology were previously investigated in Gryllus, Scapteriscus, and Velarifictorus crickets, males, and females.
A major shortcoming of this report is that while the authors suggest adding some missing information to our previous knowledge, they largely fail in presenting to the reader both the current knowledge and the potential gaps or missing pieces in it. Hence it is difficult to assess the merit and actual contribution of the current work.
Some further specific comments can be found below:
· Abstract LL25-26: This sentence is not very clear. It may be better if it read:
In crickets demonstrating only short flights, repeated flying for 30 or 60 min significantly promoted reproductive development. However, in those capable of long flights, reproductive development was affected only after a flight of 60 min.
· LL45-47: some explanatory note is required regarding the significance of white vs. pink muscles.
· LL48-68: In accord with my major point above, anecdotal information is provided on the interactions between flight and reproduction in SW or LW, males or females, in various cricket species. The reader can’t compose a consistent picture of the currently available knowledge.
· LL76-77: Does “kept separately” mean that males were separated from females? Adults separated from young? Insects were kept individually?
· LL83-84: “Flight experiments were continued for 4 h and the chamber was illuminated using a red light during the dark period.” Not clear – were experiment conducted during the day? The night? Both? (Illuminated only during the dark period?)
· LL86-87: Unclear. Do you mean that only two consecutive flight bouts were investigated (for example 5 min flight, 2 min rest, 7 min flight), before the experiment was terminated (so, why 4hr??)
· LL92-100: Not clear. Do you mean each female was prompted to fly 4 times: if all attempts ended in less than 10 min, it was depicted as a short flyer. However, if at least one attempt was longer than 20 min – a long flyer? And if all were, for example, 11-15 min long?? and if some were less than 10 min and some longer, but still less than 20 min?
· I am still unclear about the “observation period of 4 h”. I believe most experiments ended after 20-30 min.
· LL 92-100: This section appears again in the Results section, where it actually belongs.
· LL 117-120: Does this result describe short flyers, or long flyers, or both??
· LL 126-136” How many insects were tested (the data show 40, the discussion mentions 200).
· LL 134-135: What about “the maximum single flight time of some individuals were longer than 10 min but shorter than 20 min” (L 129)
· LL126-136: Could it be that Short flyers and Long flyers differ in the dynamics of the development of their flight capacity (flight capacity developing slower in Short flyers)?
· LL 140-156: in all this work it is very unclear what was the control group for each experiment.
· LL 184: “all of the 200 individuals tested (Fig. 2).” Based on Fig 2 there were ~40 insects (as ALL females flew less than 10 min in the first day).
· 225-238: Since the authors chose to discuss the control of flight behavior, it is extremely relevant to discuss potential differences in flight energy metabolism and especially a possible role of AKH (levels, release, activity) in the observed differences in flight capacity (much studied in locusts).
· LL 242-243: I believe the text is in need for yet another round of careful English proof reading.
Author Response
Interaction between flight and reproductive development is a topic that have attracted much research attention in flying animals, and specifically in insects. Some important evolutionary aspects were suggested in cases where the insect species demonstrate flight-related polymorphism (morphological or behavioral). Different cricket species have been previously reported to show wing dimorphism – i.e., short vs. long wings morphs. The interactions between this dimorphism and aspects of the insects’ reproductive biology were previously investigated in Gryllus, Scapteriscus, and Velarifictorus crickets, males, and females.
A major shortcoming of this report is that while the authors suggest adding some missing information to our previous knowledge, they largely fail in presenting to the reader both the current knowledge and the potential gaps or missing pieces in it. Hence it is difficult to assess the merit and actual contribution of the current work.
Answer: When LW females were not allowed to fly, SW females generally developed their ovaries more rapidly and laid a higher number of eggs compared with LW females, thereby suggesting a trade-off between the capability of flight and reproduction. However, LW adults will fly in the field, so it is important to investigate the effect of flight behavior on reproduction for examination of whether this trade-off exists under natural condition. So far, effect of flight behavior on reproduction of LW individuals has been rarely studied, and contradictory results are given. Therefore, we tested whether flight ability varied within LW females, and then determined the effect of flight on ovarian development in LW females with differences in terms of their flight ability in the present study.
Some further specific comments can be found below:
- Abstract LL25-26: This sentence is not very clear. It may be better if it read:
In crickets demonstrating only short flights, repeated flying for 30 or 60 min significantly promoted reproductive development. However, in those capable of long flights, reproductive development was affected only after a flight of 60 min.
Answer: it has been revised accordingly, see in lines 25-27.
- LL45-47: some explanatory note is required regarding the significance of white vs. pink muscles.
Answer: explanatory notes have been added, see in lines 47-50.
- LL48-68: In accord with my major point above, anecdotal information is provided on the interactions between flight and reproduction in SW or LW, males or females, in various cricket species. The reader can’t compose a consistent picture of the currently available knowledge.
Answer: it has been revised accordingly, see in lines 57-68.
- LL76-77: Does “kept separately” mean that males were separated from females? Adults separated from young? Insects were kept individually?
Answer: it has been revised to “individually”, see in line 85.
- LL83-84: “Flight experiments were continued for 4 h and the chamber was illuminated using a red light during the dark period.” Not clear – were experiment conducted during the day? The night? Both? (Illuminated only during the dark period?)
Answer: it has been revised to “Flight experiments were conducted during the first 4 h of night and the chamber was illuminated using a red light during the dark period”. see in lines 93-94.
- LL86-87: Unclear. Do you mean that only two consecutive flight bouts were investigated (for example 5 min flight, 2 min rest, 7 min flight), before the experiment was terminated (so, why 4hr??)
Answer: it has been revised to “flight experiments were ended when crickets did not fly in two successive tests.” see in line 97.
- LL92-100: Not clear. Do you mean each female was prompted to fly 4 times: if all attempts ended in less than 10 min, it was depicted as a short flyer. However, if at least one attempt was longer than 20 min – a long flyer? And if all were, for example, 11-15 min long?? and if some were less than 10 min and some longer, but still less than 20 min?
Answer: cricket fly less than 10 min in all 4 tests were depicted as short flyer, and cricket fly longer than 20 min at least one attempt as long flyer. Those between 10 and 20 min were not analyzed in following experiments.
- I am still unclear about the “observation period of 4 h”. I believe most experiments ended after 20-30 min.
Answer: 4 h is the longest observation period. Some crickets only fly 20-30 min, and their flight experiments were ended thereafter.
- LL 92-100: This section appears again in the Results section, where it actually belongs.
Answer: this section has been moved to the Results, see in lines 136-141.
- LL 117-120: Does this result describe short flyers, or long flyers, or both??
Answer: this result describe both short and long flyers.
- LL 126-136” How many insects were tested (the data show 40, the discussion mentions 200).
Answer: 40 crickets were tested at each day.
- LL 134-135: What about “the maximum single flight time of some individuals were longer than 10 min but shorter than 20 min” (L 129)
Answer: we did not analyze these individuals in the present study.
- LL126-136: Could it be that Short flyers and Long flyers differ in the dynamics of the development of their flight capacity (flight capacity developing slower in Short flyers)?
Answer: because the flight muscles of short flyers and long flyers were equally well developed, we do not think flight capacity develop slower in short flyer.
- LL 140-156: in all this work it is very unclear what was the control group for each experiment.
Answer: we used unflown short flyers as control when analyzing effect of flight time on ovrian development of short flyers, and unflown long flyers as control when analyzing effect of flight time on ovrian development of long flyers, see in fig.3 and fig.4.
- LL 184: “all of the 200 individuals tested (Fig. 2).” Based on Fig 2 there were ~40 insects (as ALL females flew less than 10 min in the first day).
Answer: this sentence has been revised accordingly, see in lines 184-185.
- 225-238: Since the authors chose to discuss the control of flight behavior, it is extremely relevant to discuss potential differences in flight energy metabolism and especially a possible role of AKH (levels, release, activity) in the observed differences in flight capacity (much studied in locusts).
Answer: it has been revised accordingly, see in lines 214-231.
- LL 242-243: I believe the text is in need for yet another round of careful English proof reading.
Answer: We will invite Professor Liu of Eastern Illinois University to revise the language after revision of our manuscript.
Reviewer 4 Report
Review for Ren et al. Effects of flight on reproductive development…
The authors investigate the oogenesis flight syndrome in long-winged female crickets: the propensity to fly in teneral adults and their subsequent reproductive maturation. This subject has a fairly long history – at least to C.G. Johnson who named the syndrome in 1969. My major comment is that it would be nice to know if the data analyzed with ANOVA met the assumptions for parametric tests. From the description in section 3.1 and the histograms in Fig. 2, the data do not appear to be normally distributed and they probably require some transformation (e.g., log or square root) to meet that assumption for ANOVA.
Minor comments:
l. 81 The method to investigate flight ability should be described in detail. The cited reference is not readily available – at least at my institution, and the method goes further back than Zeng 2014.t
l. 90 I suggest replacing ‘discarded’ with excluded.
Section 2.3 Shouldn’t this be moved to the Results section? The rationale for having results in the Methods section is not clear.
l. 118 Here and elsewhere change ‘AVOVA’ to ANOVA
l. 142 change ‘were’ to was
legends Fig. 3 and 4: change ‘ANOVA’ to post-hoc multiple comparisons of the means
l. 174 Here you only cite a Hemiptera example. What about Locusta migratoria and other examples. Classically I believe this was also done in milkweed bugs, Oncopeltus fasciatus, and should probably also be cited.
l. 236 The manuscript needs a stronger closing sentence.
Author Response
The authors investigate the oogenesis flight syndrome in long-winged female crickets: the propensity to fly in teneral adults and their subsequent reproductive maturation. This subject has a fairly long history – at least to C.G. Johnson who named the syndrome in 1969. My major comment is that it would be nice to know if the data analyzed with ANOVA met the assumptions for parametric tests. From the description in section 3.1 and the histograms in Fig. 2, the data do not appear to be normally distributed and they probably require some transformation (e.g., log or square root) to meet that assumption for ANOVA.
Answer: the data of fig. 1 were log transformed.
Minor comments:
- 81 The method to investigate flight ability should be described in detail. The cited reference is not readily available – at least at my institution, and the method goes further back than Zeng 2014.t
Answer: it is very simple to stimulate crickets to fly, firstly, glued crickets to a wooden stick, secondly, tied the stick with a tiny rope, thirdly, hang crickets in front of a small fan.
- 90 I suggest replacing ‘discarded’ with excluded.
Answer: this sentence has been deleted.
Section 2.3 Shouldn’t this be moved to the Results section? The rationale for having results in the Methods section is not clear.
Answer: this section has been moved to the Results, see in lines 136-141.
- 118 Here and elsewhere change ‘AVOVA’ to ANOVA
Answer: it has been revised accordingly, see in Results.
- 142 change ‘were’ to was
Answer: it has been revised, see in line 146.
legends Fig. 3 and 4: change ‘ANOVA’ to post-hoc multiple comparisons of the means
Answer: it has been revised, see in fig. 3 and fig.4
- 174 Here you only cite a Hemiptera example. What about Locusta migratoria and other examples. Classically I believe this was also done in milkweed bugs, Oncopeltus fasciatus, and should probably also be cited.
Answer: study in Oncopeltus fasciatus has been cited, see in line 177.
- 236 The manuscript needs a stronger closing sentence.
Answer: we think this closing sentence should be appropriate.
Round 2
Reviewer 2 Report
The authors followed my suggestions and revised the manuscript in most points, resulting in a very good manuscript. However, as I pointed out in the previous version, there are serious problems in the statistical methods, and the manuscript cannot be accepted until these problems are corrected. In addition, all data in this paper is only about females, so this should be kept in mind and corrected in each section.
Title
L3 female crickets
Please revise the summary and abstract according to the revised manuscript.
Introduction
Since only females were examined, the intro should also focus on females. L42-L50 include males? the Trade-off story of females is mentioned after L51, so the content is duplicated and some sentences should be deleted.
L61-64. It is hard to understand what you mean here. It just means that the trade-off is hard to measure in the field, right?
L61. LW females
M&M
L79. You are saying that you have kept the crickets for more than 10 generations, and that you have added new individuals from time to time, right? Please add information about them and the breeding density here.
L80. What kind of vegetables?
L87. What is the sample size of each experimental treatment?
L92. 2.5 m/sec. (Include reference.)
L96. In response of my following question, Why ‘2 min.’?.
Your answer: Based on our observation, a rest of 2 min is enough for crickets to fly again.
It is a totally subjective assertion and lacks objectivity. Do you have any objective data to show that two minutes is sufficient?
L100. 2.3. Effect of flight on reproductive development.
L111. In response of my question,
Your answer: we did not compare short and long flyers.
You're comparing them on L160-161.
Also, the authors should compare and discuss Long and Short, right? Then the L197 discussion will be clearer.
Please change the statistical method appropriately.
Results
L121. flight ability reached maximum at 5 days
L132-134. How many days old individuals are you talking about?
L137-140. Content duplicates L133-134.
Discussion
L171-172. Deleted because it is obvious.
L178-184. This is information that should be stated in the intro.
L186. over 4 h
L190-192. The different flight abilities of LW female V. aspersus crickets also reveal that flight behavior affects reproductive development differently.
Author Response
The authors followed my suggestions and revised the manuscript in most points, resulting in a very good manuscript. However, as I pointed out in the previous version, there are serious problems in the statistical methods, and the manuscript cannot be accepted until these problems are corrected. In addition, all data in this paper is only about females, so this should be kept in mind and corrected in each section.
Answer: Dear reviewer, thank you for your precious comments to improve our manuscript, and we are trying to do our best to revise our manuscript according to your comments. As you can see, the statistical methods has been revised, and other comments have been also revised. If you are not satisfied with our revision, we will further revise our manuscript according to your comments.
Title
L3 female crickets
Answer: it has been revised, see in L3.
Please revise the summary and abstract according to the revised manuscript.
Answer: it has been revised, see in the summary and abstract.
Introduction
Since only females were examined, the intro should also focus on females. L42-L50 include males? the Trade-off story of females is mentioned after L51, so the content is duplicated and some sentences should be deleted.
Answer: studies in L42-L50 has been revised as females.
L61-64. It is hard to understand what you mean here. It just means that the trade-off is hard to measure in the field, right?
Answer: Dear reviewer, we are trying to say LW females will fly in the field, but these previous study did not examine the effect of flight on reproduction, so we investigated the effect of flight on reproduction of LW insects.
L61. LW females
Answer: it has been revised, see in L65.
M&M
L79. You are saying that you have kept the crickets for more than 10 generations, and that you have added new individuals from time to time, right? Please add information about them and the breeding density here.
Answer: we added the sentence “We also collected the field crickets at each year and reared them with the laboratory population”, but we did not know the breeding density of these crickets in the field.
L80. What kind of vegetables?
Answer: We use the cabbages, see in L85.
L87. What is the sample size of each experimental treatment?
Answer: 40 crickets for each day, see in L94.
L92. 2.5 m/sec. (Include reference.)
Answer: We did not test the wind speed. We used 2nd gear of a fan, and it is about 2.5 m/sec according their user’s manual.
L96. In response of my following question, Why ‘2 min.’?.
Your answer: Based on our observation, a rest of 2 min is enough for crickets to fly again.
It is a totally subjective assertion and lacks objectivity. Do you have any objective data to show that two minutes is sufficient?
Answer: We just found out 2 min of rest may stimulate the cricket of this species to fly again in our preliminary observation, but we can not be sure this time is enough for other cricket species.
L100. 2.3. Effect of flight on reproductive development.
Answer: It has been revised according to this comment, see in L104.
L111. In response of my question,
Your answer: we did not compare short and long flyers.
You're comparing them on L160-161.
Also, the authors should compare and discuss Long and Short, right? Then the L197 discussion will be clearer.
Please change the statistical method appropriately.
Answer: we re-analyzed these data by muti-way ANOVA to investigate the effects of flight ability, flight time, and time after flight on flight muscle and ovarian mass, see in L115-116, and L154-159.
Results
L121. flight ability reached maximum at 5 days
Answer: it has been revised, see in L127.
L132-134. How many days old individuals are you talking about?
Answer: The days from 5-9 days when the crickets’ flight ability were strong.
L137-140. Content duplicates L133-134.
Answer: It is has been revised according to comments made other reviewers
Discussion
L171-172. Deleted because it is obvious.
Answer: It has been deleted.
L178-184. This is information that should be stated in the intro.
Answer: it has been moved to the introduction, see in L41-43.
L186. over 4 h
Answer: it has been revised, see in L193.
L190-192. The different flight abilities of LW female V. aspersus crickets also reveal that flight behavior affects reproductive development differently.
Answer: it has been revised, see in L198-199.
Reviewer 3 Report
Practically all my comments were sufficiently addresses by the authors
Author Response
this reviewer did not have comments in this round.
Reviewer 4 Report
Line 25 insert LW before ‘crickets’
Ine 29 insert LW before V. asperses
Keywords: Since the species is in the title, there’s no need to include it as a keyword. Why not use a more general keyword not found in the title, such as ‘migration’ instead.
l. 109 Did the authors conduct a test for normal distribution? Please present it.
Fig. 1 Need units on y-axis: (log10 minutes). Change ANOVA to ‘post hoc multiple comparisons of the means’.
Fig. 3 and 4 ‘comparisons’ is misspelled.
l. 147 change ‘was’ to were
l. 148 Something is missing. Do you mean ‘48 h after completion of the flight’?
l. 187 These intermittent flights over a 4 h period constitute a very loose definition of migratory. I would suggest restating ‘distances that could be considered migratory’ as ‘distances that if a consistent spatial orientation is maintained could be considered migratory’, which would be more consistent with one attribute of migratory behavior: ‘straightened-out path’.
l. 223 I really appreciate the addition of the discussion of AKH and JH hormones. However it needs a couple of topic sentences that relate the information back to the LW V. aspersus crickets, like you did for the discussion of flight fuels in line 222. Why not make a new paragraph for each hormone and tie the discussion of each hormone back to LW V. asperses.
l. 241 Can you explain how the present study has anything to do with the evolution of flightlessness? That sentence is a bit of a stretch because it sits all alone. Perhaps you could use it as a topic sentence for a final paragraph. Note that allocation of mass to reproduction relative to flight muscle alters weight loading, which affects flight speed, maneuverability, and the ability of insects to disperse. This is true both when comparisons are done among insect species, between populations of insects in transient versus more stable habitats, between insect wing morphs, and with experimental manipulation of individual insects.
Author Response
Line 25 insert LW before ‘crickets’
Answer: it has been revised, see in L25.
Ine 29 insert LW before V. asperses
Answer: it has been revised, see in L30.
Keywords: Since the species is in the title, there’s no need to include it as a keyword. Why not use a more general keyword not found in the title, such as ‘migration’ instead.
Answer: it has been revised, see in L31.
- 109 Did the authors conduct a test for normal distribution? Please present it.
Answer: it has been revised, see in L112-113. The normal distribution of flight duration has been tested by K-S test, P=0.008, suggesting the data were not normally distributed.
Fig. 1 Need units on y-axis: (log10 minutes). Change ANOVA to ‘post hoc multiple comparisons of the means’.
Answer: it has been revised, see in fig.1.
Fig. 3 and 4 ‘comparisons’ is misspelled.
Answer: it has been revised, see in fig.3 and 4.
- 147 change ‘was’ to were
Answer: it has been revised, see in L160.
- 148 Something is missing. Do you mean ‘48 h after completion of the flight’?
Answer: it has been revised, see in L161.
- 187 These intermittent flights over a 4 h period constitute a very loose definition of migratory. I would suggest restating ‘distances that could be considered migratory’ as ‘distances that if a consistent spatial orientation is maintained could be considered migratory’, which would be more consistent with one attribute of migratory behavior: ‘straightened-out path’.
Answer: it has been revised, see in L194-195.
- 223 I really appreciate the addition of the discussion of AKH and JH hormones. However it needs a couple of topic sentences that relate the information back to the LW V. aspersus crickets, like you did for the discussion of flight fuels in line 222. Why not make a new paragraph for each hormone and tie the discussion of each hormone back to LW V. asperses.
Answer: it has been revised, see in L245-251.
- 241 Can you explain how the present study has anything to do with the evolution of flightlessness? That sentence is a bit of a stretch because it sits all alone. Perhaps you could use it as a topic sentence for a final paragraph. Note that allocation of mass to reproduction relative to flight muscle alters weight loading, which affects flight speed, maneuverability, and the ability of insects to disperse. This is true both when comparisons are done among insect species, between populations of insects in transient versus more stable habitats, between insect wing morphs, and with experimental manipulation of individual insects.
Answer: this sentence has been deleted.
Round 3
Reviewer 2 Report
L83. with the laboratory population for more than 10 generations.
In addition, please indicate how many individuals you started breeding and how many you are maintaining. This is because the degree of genetic diversity depends on the initial number of reared individuals, and in some cases, founder effects and other factors must be discussed.
L100. I am still wondering why 2 minutes. I think it would be better to have a summary statement following. As a personal observation, I think the following sentence should be summarized and stated. 'We just found out 2 min of rest may stimulate the cricket of this species to fly again in our preliminary observation'.
L135. Figures 3 and 4 should be combined since they are compared with Short fliers and Long fliers.
Author Response
Dear reviewer, thank you for your precious comments, and we have revised our manuscript according to these comments. If you are not satisfied with our revision, we will further revise our manuscript. Thanks again.
L83. with the laboratory population for more than 10 generations.
In addition, please indicate how many individuals you started breeding and how many you are maintaining. This is because the degree of genetic diversity depends on the initial number of reared individuals, and in some cases, founder effects and other factors must be discussed.
Answer: We started the lab population from dozens of crickets, and we maintained about 100 adults for each generation, see in L 83-84.
L100. I am still wondering why 2 minutes. I think it would be better to have a summary statement following. As a personal observation, I think the following sentence should be summarized and stated. 'We just found out 2 min of rest may stimulate the cricket of this species to fly again in our preliminary observation'.
Answer: It has been revised according to this comment, see in L100-101.
L135. Figures 3 and 4 should be combined since they are compared with Short fliers and Long fliers.
Answer: The fig.3 and fig.4 were combined, see in fig.3.